# The BETTY Score to Predict Perioperative Outcomes in Surgical Patients

**DOI:** 10.3390/cancers15113050

**Published:** 2023-06-04

**Authors:** Michael Baboudjian, Rawad Abou-Zahr, Bogdan Buhas, Alae Touzani, Jean-Baptiste Beauval, Guillaume Ploussard

**Affiliations:** 1Department of Urology, La Croix du Sud Hôpital, 31130 Quint Fonsegrives, France; rawad.abouzahr@gmail.com (R.A.-Z.); medalaet@gmail.com (A.T.); jbbeauval@gmail.com (J.-B.B.); g.ploussard@gmail.com (G.P.); 2Department of Urology, North Hospital, Aix-Marseille University, Assistance Publique des Hôpitaux de Marseille (APHM), 13007 Marseille, France

**Keywords:** surgery, complication, score, BETTY, radical prostatectomy

## Abstract

**Simple Summary:**

Careful monitoring and analysis of surgical outcomes is crucial for ensuring the safety and quality of clinical care. Therefore, a simple and low-cost metric of the risk of postoperative adverse events that would provide timely feedback to surgical teams in any setting is needed. However, current models primarily include anesthetic data that are not readily available to surgeons, and their ability to predict surgical outcomes has been often questioned. Here, we proposed a new user-friendly scoring system, namely the BETTY score and found that it was strongly associated with postoperative morbidity after radical prostatectomy. Future studies, in various surgical subspecialties, are ongoing to confirm the usefulness of this easy-to-use score in routine.

**Abstract:**

The aim of this study is to evaluate a new user-friendly scoring system, namely the BETTY score, that aims to predict 30-day patient outcomes after surgery. In this first description, we rely on a population of prostate cancer patients undergoing robot-assisted radical prostatectomy. The BETTY score includes the patient’s American Society of Anesthesiologists score, the body mass index, and intraoperative data, including operative time, estimated blood loss, any major intraoperative complications, hemodynamic, and/or respiratory instability. There is an inverse relationship between the score and severity. Three clusters assessing the risk of postoperative events were defined: low, intermediate, and high risk of postoperative events. A total of 297 patients was included. The median length of hospital stay was 1 day (IQR1-2). Unplanned visits, readmissions, any complications, and serious complications occurred in 17.2%, 11.8%, 28.3%, and 5% of cases, respectively. We found a statistically significant correlation between the BETTY score and all endpoints analyzed (all *p* ≤ 0.01). A total of 275, 20, and 2 patients were classified as low-, intermediate-, and high-risk according to the BETTY scoring system, respectively. Compared with low-risk patients, patients at intermediate-risk were associated with worse outcomes for all endpoints analyzed (all *p* ≤ 0.04). Future studies, in various surgical subspecialties, are ongoing to confirm the usefulness of this easy-to-use score in routine.

## 1. Introduction

Postoperative complications are common, up to 40% of general surgery cases [1], and have been associated with prolonged hospitalization, increased risk of readmission, and mortality, with significant cost implications [2,3]. In the United States, an estimated 19.5% of all Medicare beneficiaries discharged from the hospital in 2003 were readmitted within 30 days, resulting in a cost of USD 17.4 billion. As a result, surgical complications quickly became an important metric for measuring the quality of patient care. Individualized surgical risk prediction tools represent a unique opportunity for shared decision making in surgical patients and for adapting the surveillance (at hospital and after discharge) burden during the postoperative course. The current challenges are to correctly identify high-risk patients who should benefit from more extensive surveillance protocols in order to identify complications early and/or to improve their prevention. In this regard, several scores/calculators have been developed that rely on preoperative variables (e.g., the Veterans Affairs Surgical Quality Improvement Program [4] or American College of Surgeons National Surgical Quality Improvement Program (ACS NSQIP) [5,6,7]) but they do not consider the magnitude of surgery or intraoperative events. Thus, surgeons still rely on the subjective assessment of overall patient conditions after surgical procedures for clinical communication, postoperative triage, and decision making. In 2007, Gawande et al. proposed a 10-point Surgical Apgar Score for predicting major postoperative complications and mortality, based on intraoperative parameters [8]. Although this score is currently the most widely used, it was only designed to predict a certain type of event (i.e., major morbidity/mortality), does not take into account the preoperative health status of the patient, and is not applicable to all surgeries.

Herein, we sought to introduce a new user-friendly scoring system, namely the BETTY score, which is based on preoperative and intraoperative parameters, and which aims to predict a wide range of postoperative events. In this first description, we applied the BETTY score to a population of prostate cancer (PCa) patients undergoing robot-assisted radical prostatectomy (RARP).

## 2. Materials and Methods

### 2.1. Patients

Patients were identified from a prospectively maintained single institution database and were recruited between 2018 and 2022. This study was conducted in accordance with the principles of Good Clinical Practice and the Declaration of Helsinki. All data were deidentified and the study protocol was approved by the local institutional ethics committee (IRB number: 0010835). The cohort included all patients aged ≥ 18 years with histologically confirmed, clinically localized PCa. All patients underwent systematic and MRI-targeted biopsies and subsequent RP with or without pelvic lymph node dissection. The decision to perform RP was left to the clinical judgment of the treating physician after discussion with each patient regarding the potential benefits and side effects of all available treatment modalities for the management of PCa [9].

### 2.2. Procedures

RP ± pelvic lymph node dissection was performed using a transperitoneal robotic-assisted approach (RARP). RARP was performed by five experienced surgeons, all beyond their learning curve, having performed more than 200 procedures at study entry. The surgical technique has been widely described previously [10]. No variation in surgical technique (standard transperitoneal approach, nerve-sparing, apex reconstruction, extent of lymph node dissection, and bladder neck sparing) was noted among the surgeons during the study period. The postoperative course was standardized in terms of care (thromboprophylaxis for three weeks, analgesics on demand). The bladder catheter was removed on day 7. The first postoperative visit was scheduled after one month for all patients.

### 2.3. The BETTY Score

The newly developed and user-friendly BETTY score includes six variables presumably associated with surgical outcomes. The variables to be included in the score were discussed among the authors on the basis of a literature review, and final approval of the score was obtained when a consensus was reached among all authors. The score is intended to be applied to any surgical procedure and available on a mobile app (www.betty.care, accessed on 5 April 2023). Prospective data collection in various surgical settings is ongoing to evolve the score through a machine learning process. Therefore, this score is subject to change over time.

The BETTY score includes preoperative data, including the patient’s American Society of Anesthesiologists (ASA) score and body mass index, as well as intraoperative data, including operative time, estimated blood loss, any major intraoperative complications, hemodynamic instability, and/or respiratory instability. Table 1 provides an overview of the scoring system. There is an inverse relationship between the BETTY score and severity (i.e., when the score increases, severity decreases). Three clusters assessing the risk of postoperative events were a priori defined: low risk, intermediate risk, and high risk.

### 2.4. Endpoints

The new BETTY score aims to predict the postoperative outcome of patients, and we identified five short-term endpoints for patients undergoing RARP: any 30-day postoperative complication; 30-day high-grade complication defined as a Clavien–Dindo event ≥ 3 according to the Clavien–Dindo classification system [11], length of hospital stay, unplanned patient visit (i.e., emergency department visit), and unplanned readmission. 

### 2.5. Data Analysis

Descriptive statistics were carried out with the available variables. Categorical variables were reported as frequencies and percentages (%), and continuous variables as medians and interquartile ranges (IQRs). Statistical analyses were performed in two steps. First, Spearman and Pearson correlation analyses were performed to clarify the relationship between the BETTY score and the five predefined postoperative assessment criteria. Subsequently, multivariable logistic regression models were used to assess the association between the BETTY score and all of the assessment criteria described above. Models were adjusted for patient age, Charlson comorbidity index, and use of antithrombotic agents. All statistical analyses were performed using R software Version 4.1.3 (R Foundation for Statistical Computing, Vienna, Austria). All tests were two-sided, with significance level set at *p* < 0.05.

## 3. Results

A total of 297 consecutive patients met our inclusion criteria. Baseline characteristics are summarized in Table 2. The median patient age was 67 years (IQR 62–71), most of patients had an ASA 1 score (95.3%), and the median patient body mass index was 26 (IQR 24–28). The median preoperative PSA value was 7.1 ng/mL (IQR 5.4–10) and 287 patients (96.6%) had clinically significant PCa (i.e., Gleason grade Group ≥ 2).

Table 3 describes perioperative outcomes. The estimated median blood loss was 200 cc (IQR 150–350), ten cases (3.4%) of intraoperative hemodynamic instability were noted, and only one patient required a blood transfusion. Pelvic lymph node dissection was performed in 232 cases (78.1%), and the median operating time (skin-to-skin) was 149 min (IQR 126–174). There was no conversion to open surgery. The median length of hospital stay was 1 day (IQR 1–2). Unplanned visits, readmissions, any complications, and serious complications occurred in 17.2%, 11.8%, 28.3%, and 5% of cases, respectively.

As shown in Table 4, we found a statistically significant correlation between the BETTY score and all endpoints analyzed, including continuous (i.e., length of hospital stay: correlation coefficient [r] = −0.27, *p* < 0.001) and categorical parameters (i.e., any complication: r = −0.13, *p* = 0.01; high-grade complication: r = −0.28, *p* < 0.001; unplanned visit: r = −0.13, *p* = 0.01; unplanned readmission: r = −0.14, *p* = 0.01). The direction of the association was as expected: as the score increased (meaning that severity decreased), the risk of postoperative adverse events decreased.

A total of 275, 20, and 2 patients were classified as low-, intermediate-, and high-risk, according to the BETTY scoring system, respectively. As shown in Table 5, the risk of postoperative events (i.e., any complication, high-grade complication, unplanned visit, and readmission) and length of hospital stay progressively increased from low- to high-risk patients. However, due to the low number of patients included in the high-risk group, these patients were excluded from the multivariate models. Compared with low-risk patients, patients at intermediate-risk were associated with an increased risk of any grade complication (adjusted OR 5.4, 95% CI 2 to 15, *p* < 0.001), high-grade complication (adjusted OR 16, 95% CI 4.7 to 56, *p* < 0.001), longer hospital stay (adjusted OR 2.72, 95% CI 1.01 to 6.97, *p* = 0.04), unplanned visit (adjusted OR 3.5, 95% CI 1.3 to 9, *p* = 0.01) and unplanned readmission (adjusted OR 6, 95% CI 2.1 to 16, *p* < 0.001).

Models were adjusted for patient age, Charlson comorbidity index, and anti-thrombotic agents use. The anti-thrombotic agents use was significantly associated with any complication (OR 2.22, 95% CI 1.14 to 4.29, *p* = 0.02) and unplanned readmission (OR 2.50, 95% CI 1.03 to 5.77, *p* = 0.03). Patient age and Charlson comorbidity index were not associated with all analyzed endpoints (all *p* > 0.05).

## 4. Discussion

Careful monitoring and analysis of surgical outcomes is crucial for ensuring the safety and quality of clinical care, counseling patients, and conducting public health research. Therefore, a simple and low-cost metric of the risk of postoperative adverse events that would provide timely feedback to surgical teams in any setting is needed. In addition, postoperative adjuvant therapy is often offered, and the score could influence the indications and timing of these therapies. However, current models primarily include anesthetic data that are not readily available to surgeons or focus solely on the preoperative status of the patient. As a result, their ability to predict surgical outcomes has often been questioned. To address this void, we sought to introduce a new user-friendly scoring system, namely the BETTY score, and to evaluate its ability to predict the postoperative course of patients after surgery. In this first description, we applied the BETTY score to a population of PCa patients undergoing RARP. Our study revealed several noteworthy findings.

We found that the BETTY score was strongly associated with postoperative morbidity following RARP in PCa patients. This association remained unchanged when we adjusted our analysis for patient age, Charlson comorbidity index, and use of antithrombotic agents, which are known to influence postoperative patient outcomes after RARP [12,13,14,15,16,17]. As a result, the BETTY score passed the first test, and these preliminary results support the development and refinement of the new scoring system in various surgical settings.

RP is one of the main options to treat localized PCa and is usually offered to young and healthy patients. Over the past two decades, several changes in the surgical technique of RP have been implemented in our practices, the most well-known being the introduction of the robotic approach to achieve an additional level of precision during surgery [18,19]. As a result, we have recently demonstrated that the surgical approach is a key element influencing the perioperative outcomes of RP [20,21]. Among 19,018 RPs performed in France in 2020, we found that RARP was associated with lower complication rates, shorter length of stay, and lower readmission rates, compared to open and laparoscopic approaches [21]. Thus, assessing the value of the BETTY score in a population of young patients treated with safe surgery was challenging, and explains the very low number of patients defined as high risk in this study. Therefore, we could not include this subgroup in our multivariate analyses, which may be considered a drawback. However, all patients classified as high risk experienced postoperative event, suggesting the clinical interest to identify that sub-population of fragile patients for guiding the postoperative course and anticipating potential complications. The interest of this new score is that it is applicable to all types of surgery, and future studies should evaluate whether the subgroup of high-risk patients is indeed, or not, associated with the highest postoperative morbidity.

The 10-point surgical Apgar score is intended to provide an objective, immediate, and easily calculated summary assessment of a patient’s condition after surgery, to identify patients at high risk for major complications and to provide an objective summary for communication between different teams. The components of the score are estimated blood loss, lowest mean arterial pressure, and lowest heart rate during surgery, reflecting intraoperative hemodynamics [5]. Despite its simplicity, the application to surgery of the Apgar score, which was first developed for newborns [22], poses some problems. First, the data collected in the Apgar score include only intraoperative parameters and omit the patient’s preoperative status, which plays a major role in postoperative recovery, as well as the contemporary development of minimally invasive surgery, which leads to the necessary need for reconsidering initially proposed thresholds of blood loss. Second, the anesthesia record should include heart rate and blood pressure measurements at acceptable intervals, and these data should be available to clinicians/surgeons at any time, even retrospectively. Third, the Apgar score was developed to predict only major postoperative complications and does not predict other adverse events, such as unplanned readmission, minor complications, or prolonged hospitalization, which can compromise patients’ health-related quality of life and increase health care costs. Fourth, the Apgar score does not appear to be appropriate in surgical settings with low risk of postoperative adverse events or in surgeries performed under regional anesthesia, with several studies that reported a moderate discriminatory ability in various surgical subspecialities [23,24,25,26,27]. In these studies, the authors generally found that the preoperative functional status was a more important predictor than the Apgar score for the occurrence of adverse postoperative events [28]. The new BETTY score has the potential to overcome these drawbacks due to its features: the inclusion of preoperative and intraoperative parameters, easy-to-access and use data, and broad applicability across surgical subspecialties. We are currently developing a large prospective data collection in various surgical specialties, which will allow us to validate the BETTY score on a large scale. In addition, the data will be analyzed by machine learning, which will improve the discriminatory ability of the score over time. After external validation of our score in different surgical settings, a direct comparison of the BETTY score with currently available scores should be performed, to assess which one best evaluates the patient’s postoperative course.

The present study has some limitations that should be acknowledged. First, the main limitation lies in its single-center and retrospective design, even if data collection was performed prospectively. Second, only a small proportion of patients at high-risk were available in our dataset, which prevented us from analyzing them as a distinct subset in our multivariate model. Third, the initial description of the BETTY score was developed on the basis of a literature-based consensus that does not meet current standards for developing a scoring system. However, prospective validation of the score using a mobile app is underway and a machine learning system that collects data is built into the system to evolve the score over time. Finally, no direct comparison with the Apgar score was performed due to the lack of specific data available in our dataset (i.e., lowest mean arterial pressure and lowest heart rate during surgery, which are variables included in the Apgar score), and future studies are needed to assess whether the BETTY score may provide better discriminatory ability than the Apgar score.

## 5. Conclusions

We proposed a new user-friendly scoring system, namely the BETTY score, which aims to predict perioperative morbidity in various surgical setting. In this first description, we applied the BETTY score to a population of PCa patients undergoing RARP and found that it was strongly associated with postoperative morbidity following the major oncological surgery. Future studies, in various surgical subspecialties, are ongoing.

## Figures and Tables

**Table 1 cancers-15-03050-t001:** The BETTY score.

Variables (BETTY)	
Body mass index (kg/m^2^)	
Estimated blood loss (mL)	
Operative Time (min)	
FiTness for surgery (i.e., ASA score)	
Instability	Hemodynamic/respiratory instability during surgery
	Major intraoperative complication

Legend: ASA: American Society of Anesthesiologists.

**Table 2 cancers-15-03050-t002:** Baseline characteristics.

Variables	Overall Cohort (*n* = 297)
Age, years	67 (62–71)
BMI, kg/m^2^	26 (24–28)
Charlson Comorbidity Index	4 (4–5)
ASA score	
1	283 (95.3)
2	12 (4)
3	2 (0.7)
Anti-platelet therapy	42 (14.1)
Anticoagulant therapy	15 (5)
T stage	
T1	146 (49.2)
T2	131 (44.1)
T3	20 (6.7)
Preoperative PSA value, ng/mL	7.1 (5.4–10)
Prostate volume, mL	48 (37–63)
PSA density	0.15 (0.11–0.21)
Gleason score (GS)	
GS 6	10 (3.4)
GS 7	253 (85.2)
GS 8	30 (10.1)
GS 9	4 (1.3)

Legend: BMI: body mass index; ASA: American Society of Anesthesiologists; PSA: prostate-specific antigen. Data are presented as median (interquartile range) or number (percentage).

**Table 3 cancers-15-03050-t003:** Perioperative outcomes.

Variables	Overall Cohort (*n* = 297)
Intraoperative data	
Pelvic lymph node dissection	232 (78.1)
Estimated blood loss, mL	200 (150–350)
Blood transfusion	1 (0.3)
Major complication	5 (1.7)
Hemodynamic/respiratory instability	10 (3.4)
Operative time, min	149 (126–174)
Postoperative data	
Length of stay, days	1 (1–2)
Unplanned patient visit	51 (17.2)
Unplanned readmission	35 (11.8)
30-d complications	
Any	84 (28.3)
High grade (i.e., CD ≥ 3)	15 (5)

Legend: CD: Clavien–Dindo. Data are presented as median (interquartile range) or number (percentage).

**Table 4 cancers-15-03050-t004:** Correlation between BETTY score and postoperative outcomes.

	r	*p* Value
Any complication	−0.13	0.01
High-grade complication (CD ≥ 3)	−0.28	<0.001
Length of hospital stay	−0.27	<0.001
Unplanned visit	−0.13	0.01
Unplanned readmission	−0.14	0.01

Legend: CD: Clavien–Dindo. Correlations between BETTY score and postoperative outcomes were performed using Spearman correlation for categorical data and Pearson correlation for continuous data.

**Table 5 cancers-15-03050-t005:** Multivariate regression analysis using the three risk groups of the BETTY score to predict postoperative outcomes.

		Low Risk BETTY ≥ 12 (*n* = 275)	Intermediate Risk BETTY 7–11 (*n* = 20)	High Risk BETTY ≤ 6 (*n* = 2)
Any complication	Rate (%)	25%	65%	100%
	Adjusted OR (95% CI)	Ref.	5.3 (2–15)	N/A
	*p*	Ref.	<0.001	N/A
High-grade complication (CD ≥ 3)	Rate (%)	2.5%	30%	100%
	Adjusted OR (95% CI)	Ref.	16 (4.7–56)	N/A
	*p*	Ref.	<0.001	N/A
Length of hospital stay, days	Median	1	1.5	1.5
	Adjusted OR (95% CI)	Ref.	2.72 (1.01–6.97)	N/A
	*p*	Ref.	0.04	N/A
Unplanned visit	Rate (%)	14.9%	40%	100%
	Adjusted OR (95% CI)	Ref.	3.5 (1.3–9)	N/A
	*p*	Ref.	0.01	N/A
Unplanned readmission	Rate (%)	9%	40%	100%
	Adjusted OR (95% CI)	Ref.	6 (2.1–16)	N/A
	*p*	Ref.	<0.001	N/A

Legend: OR: odds ratio; CI: confidence interval; CD: Clavien–Dindo; N/A: not available.

## Data Availability

Data are available upon reasonable request to the corresponding author.

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
