# Peer review of "The BETTY Score to Predict Perioperative Outcomes in Surgical Patients"

_cancers, 2023, doi:10.3390/cancers15113050_

Round 1

Reviewer 1 Report

The authors report on the usefulness of the BETTY score for perioperative complications in robot-assisted laparoscopic radical prostatectomy (RARP) using a simpler scoring system, the BETTY score which does not require anesthetic data.

One of the strengths of the BETTY score is that, unlike other scoring systems used to assess perioperative complications, it does not use anesthesia data, making it easier to assess perioperative risk in general practice.

One of the limitations is that the number of cases classified as high-risk in this BETTY score is so small in this cohort that multivariate analysis could not be performed.

The manuscript is well written scientifically and logically, contrasting with existing perioperative scoring systems. However, the following points require further consideration.

Major points:

#1. Although it is good to show that the BETTY score is associated with postoperative complications, there is a lack of consideration of how this can be returned to patients in actual clinical practice. Especially in prostate cancer, evidence-based treatments such as radiotherapy in addition to surgery have already been established, and it is necessary to thoughtfully consider the treatment strategy when the BETTY score is high.

#2. The authors state that the surgical procedure was performed by 5 surgeons who were skilled and above the learning curve, but this was a single-center study. I think the fact that it was a single-center study should be added to the limitations if standardization by BETTY score is to be achieved.

Minor points:

#1. The authors seem to have listed Gleason grade groups as variables in Table 2, while the corresponding Grades are listed in Gleason sums. Please check it again and unify it with one of them.

Author Response

Reviewer #1:

Comment 1: The authors report on the usefulness of the BETTY score for perioperative complications in robot-assisted laparoscopic radical prostatectomy (RARP) using a simpler scoring system, the BETTY score which does not require anesthetic data. One of the strengths of the BETTY score is that, unlike other scoring systems used to assess perioperative complications, it does not use anesthesia data, making it easier to assess perioperative risk in general practice. One of the limitations is that the number of cases classified as high-risk in this BETTY score is so small in this cohort that multivariate analysis could not be performed. The manuscript is well written scientifically and logically, contrasting with existing perioperative scoring systems. However, the following points require further consideration.

Response 1: We sincerely thank the reviewer for all the comments she/he made that really improved the quality of the manuscript. We provide a point-by-point response below to respond to all comments raised.

Major points:

Comment 2: Although it is good to show that the BETTY score is associated with postoperative complications, there is a lack of consideration of how this can be returned to patients in actual clinical practice. Especially in prostate cancer, evidence-based treatments such as radiotherapy in addition to surgery have already been established, and it is necessary to thoughtfully consider the treatment strategy when the BETTY score is high.

Response 2: The reviewer raised an important point. We totally agree with his/her point of view. We have added this statement in the discussion section as follows:

“In addition, postoperative adjuvant therapy is often offered, and the score could influence the indications and timing of these therapies.”

Comment 3: The authors state that the surgical procedure was performed by 5 surgeons who were skilled and above the learning curve, but this was a single-center study. I think the fact that it was a single-center study should be added to the limitations if standardization by BETTY score is to be achieved.

Response 3: The reviewer raised an important point, and we agree that the single-center design is a limitation that needs to be recognized. Accordingly, we have added this limitation in our discussion section as follows:

 “First, the main limitation lies in its single-center and retrospective design”

Minor points:

Comment 4: The authors seem to have listed Gleason grade groups as variables in Table 2, while the corresponding Grades are listed in Gleason sums. Please check it again and unify it with one of them.

Response 4: The reviewer is correct. We have kept the term "Gleason score" to avoid confusion.

Reviewer 2 Report

The authors developed and tested a simple and easy-to-use score to predict surgical outcomes, encompassing both pre-operative and intra-operative variables.

Their effort should be commended because this kind of tool is useful to adapt post-op surveillance based on event risk and their results are encouraging.

The article is well-written and the methodology is generally sound, with the limitations being discussed appropriately.

Only a few minor observations:

intro

> consider citing the score ACS NSQIP (PMID: 24055383) for comparison

methods

> l.101: on what basis were the risk clusters for the score defined a priori? please detail

> l.119: prefer "multivariable" to “multivariate” in this case, as the outcome variable is one in each of the 4 models

> consider discussing why AGE (in particular) , CCI and antithrombitc agents are used as covariates but are not included in the BETTY score.

 results

> please detail how many patients in the specified time span 2018-2022 were exclude from analysis (generalizability issues?) and why.

> please confirm if other regressors of table 5 were only AGE, CCI and Antithrombotic agents and if they were significant or not in the model (no need to modify the table).

 discussion

> consider discussing how risk subgroups were decided a priori and if they could theoretically vary according to the type of surgery.

> consider discussing some other scores (e.g. ACS NSQIP) differences, even if a formal comparison is not yet available.

> discuss how to move from regression to prediction (a nomogram?) and "external" validation.

abstract: I would add "worse outcomes" to the sentence "patients at intermediate risk were associated with [worse outcomes] for all endpoints analysed".

intro: line 42: billion--> billions

Author Response

Comment 5: The authors developed and tested a simple and easy-to-use score to predict surgical outcomes, encompassing both pre-operative and intra-operative variables. Their effort should be commended because this kind of tool is useful to adapt post-op surveillance based on event risk and their results are encouraging. The article is well-written and the methodology is generally sound, with the limitations being discussed appropriately.

Response 5: We sincerely thank the reviewer for his/her positive feedback.

Only a few minor observations:

intro

Comment 6: consider citing the score ACS NSQIP (PMID: 24055383) for comparison

Response 6: We thank the reviewer for the reference. As suggested, we added this reference in the introduction section.

methods

Comment 7: l.101: on what basis were the risk clusters for the score defined a priori? please detail

Response 7: We thank the reviewer for his/her comment. The variables to be included in the model were based on a consensus among the co-authors after a review of the literature. This could be interpreted as a limitation which we acknowledge in the discussion section as follows:

“Third, the initial description of the BETTY score was developed on the basis of a literature-based consensus that does not meet current standards for developing a scoring system.”

Comment 8:  l.119: prefer "multivariable" to “multivariate” in this case, as the outcome variable is one in each of the 4 models

Response 8: We would like to thank the reviewer for his/her careful reading. The word has been changed according to the reviewer’s comment.

Comment 9: consider discussing why AGE (in particular) , CCI and antithrombitc agents are used as covariates but are not included in the BETTY score.

Response 9: We thank the reviewer for his/her comment. The variables to be included in the model were based on a consensus among the co-authors after a review of the literature. This could be interpreted as a limitation which we acknowledge in the discussion section as follows:

“Third, the initial description of the BETTY score was developed on the basis of a literature-based consensus that does not meet current standards for developing a scoring system.”

 results

Comment 10: please detail how many patients in the specified time span 2018-2022 were exclude from analysis (generalizability issues?) and why.

Response 10: We thank the reviewer for his/her comment. In fact, we included all patients who underwent RP during the study period and did not exclude any of them. We have added the term "consecutive" to be clearer to readers.

Comment 11: please confirm if other regressors of table 5 were only AGE, CCI and Antithrombotic agents and if they were significant or not in the model (no need to modify the table).

Response 11: The reviewer raised an important point. We confirm that these 3 variables (and only these) were included as covariates. As suggested by the reviewer, we have added in the legend to Table 5 the results of the multivariate analyses with these variables, as follows:

“Anti-thrombotic agents use was significantly associated with any complication (OR 2.22, 95% CI 1.14 to 4.29, p=0.02) and unplanned readmission (OR 2.50, 95% CI 1.03 to 5.77, p=0.03).

Patient age and Charlson comorbidity index were not associated with all analyzed endpoints (all p>0.05).”

 discussion

Comment 12: consider discussing how risk subgroups were decided a priori and if they could theoretically vary according to the type of surgery.

Response 12: The reviewer raised an important point. The variables and risk groups were defined a priori based on consensus among the authors after a review of the literature. Our initial version includes a model that may evolve over time as we collect data prospectively in various surgical subspecialties. We state it as follows (discussion section):

“We are currently developing a large prospective data collection in various surgical specialties, which will allow us to validate the BETTY score on a large scale. In addition, the data will be analyzed by machine learning, which will improve the discriminatory ability of the score over time.”

“However, prospective validation of the score using a mobile app is underway and a machine learning system that collects data is built into the system to evolve the score over time.”

Comment 13: consider discussing some other scores (e.g. ACS NSQIP) differences, even if a formal comparison is not yet available.

Response 13: We thank the reviewer for his comment. Indeed, no direct comparison can be drawn from our series, but we provide throughout the manuscript the potential advantages of our score over those currently available:

Discussion section: “However, current models primarily include anesthetic data that are not readily available to surgeons or focus solely on the preoperative status of the patient. As a result, their ability to predict surgical outcomes has often been questioned. »

“The new BETTY score has the potential to overcome these drawbacks: inclusion of preoperative and intraoperative parameters, easy-to-access and use data, and broad applicability across surgical subspecialties.”

“After external validation of our score in different surgical settings, a direct comparison of the BETTY score with currently available scores should be performed to assess which one best evaluates the patient's postoperative course.”

Comment 14: discuss how to move from regression to prediction (a nomogram?) and "external" validation.

Response 14: We agree with the reviewer’s comment and we have added this very important information in the Discussion section as follows:

“After external validation of our score in different surgical settings, a direct comparison of the BETTY score with currently available scores should be performed to assess which one best evaluates the patient's postoperative course. »

Comments on the Quality of English Language

Comment 15: abstract: I would add "worse outcomes" to the sentence "patients at intermediate risk were associated with [worse outcomes] for all endpoints analysed".

Response 15: We would like to thank the reviewer for his/her careful reading. The sentence has been changed according to the reviewer’s comment.

Comment 16: intro: line 42: billion--> billions

Response 16: We apologize for this typo. The change has been made accordingly.

Reviewer 3 Report

It is an excellent study presenting a new application to predict postoperative complication risk. The study is very well designed and the results were described in an excellent way. An image of the application or the way that it works would be beneficial

Author Response

Comment 17: It is an excellent study presenting a new application to predict postoperative complication risk. The study is very well designed and the results were described in an excellent way. An image of the application or the way that it works would be beneficial

Response 17: We would like to thank the reviewer for his/her positive feedback, we really appreciate it. The BETTY score is now available online and on smartphones (https://betty.care/appstores). The link is provided in the Methods section and we would like readers to use the web application directly to test how it works.